

# The common personal behavior and preventive measures among 42 uninfected travelers from the Hubei province, China during COVID-19 outbreak: a cross-sectional survey in Macao SAR, China

Chon Fu Lio[1,*], Hou Hon Cheong[1,*], Chin Ion Lei[2], Iek Long Lo[3], Lan Yao[4], Chong Lam[5] and Iek Hou Leong[5]

[1] Macao Academy of Medicine, Health Bureau, Macao SAR, China
[2] Department of Internal Medicine, Centro Hospitalar Conde de São Januário, Health Bureau, Macao SAR, China
[3] Department of Respiratory Medicine, Centro Hospitalar Conde de São Januário, Health Bureau, Macao SAR, China
[4] Health Bureau, Macao SAR, China
[5] Center for Disease Control and Prevention, Health Bureau, Macao SAR, China
* These authors contributed equally to this work.

Corresponding author
Chin Ion Lei, cilei@ssm.gov.mo

## ABSTRACT

**Background:** The novel coronavirus diseases 2019 (COVID-19) caused over 1.7 million confirmed cases and cumulative mortality up to over 110,000 deaths worldwide as of 14 April 2020. A total of 57 Macao citizens were obligated to stay in Hubei province, China, where the highest COVID-19 prevalence was noted in the country and a "lockdown" policy was implemented for outbreak control for more than one month. They were escorted from Wuhan City to Macao via a chartered airplane organized by Macao SAR government and received quarantine for 14 days with none of the individual being diagnosed with COVID-19 by serial RNA tests from the nasopharyngeal specimens and sera antibodies. It was crucial to identify common characteristics among these 57 uninfected individuals.

**Methods:** A questionnaire survey was conducted to extract information such as behavior, change of habits and preventive measures.

**Results:** A total of 42 effective questionnaires were analyzed after exclusion of 14 infants and children with age under fifteen as ineligible for the survey and missing of one questionnaire, with a response rate of 97.7% (42 out of 43). The proportion of female composed more than 70% of this group of returners. The main reason for visiting Hubei in 88.1% of respondents was to visit relatives. Over 88% of respondents did not participate in high-risk activities due to mobility restriction. All (100%) denied contact with suspected or confirmed COVID-19 cases. Comparison of personal hygiene habits before and during disease outbreak showed a significant increase in practice including wearing a mask when outdoor (16.7% and 95.2%, $P < 0.001$) and often wash hands with soap or liquid soap (85.7% and 100%, $P = 0.031$).

# INTRODUCTION

The novel coronavirus diseases 2019 (COVID-19) caused by the severe acute respiratory syndrome coronavirus 2 (SARS-CoV-2) caused over 1.7 million confirmed cases and cumulative mortality up to over 110,000 deaths worldwide as of 14 April 2020 (*World Health Organization (WHO), 2020b*), provided with its early transmission dynamic of human–human transmission among close contacts (*Li et al., 2020*). It is estimated in a model that COVID-19 would have resulted in 7.0 billion infections and 40 million deaths globally in 2020 in the absence of any intervention (*Walker et al., 2020*). Wuhan City, the capital of Hubei province in China, became the first outbreak center of COVID-19 since December 2019 (*Phelan, Katz & Gostin, 2020*). During the Chinese New Year Holidays, Chinese people have the traditional habit of traveling to their hometowns for a family reunion and gathering to celebrate the beginning of the Lunar New Year. Hence, many people, including groups of Macao citizens, were obligated to stay in Hubei province after the announcement of "lockdown"/sanitary cordons by the local government on 23 January 2020, that is, 2 days before Chinese New Year. It was not until 7 March 2020 that the Macao SAR government escorted a special team to Wuhan, China to pick up 57 Macao citizens from 31 families, who stayed in 10 different cities in Hubei province (*Macao SAR Government Portal, 2020b*). COVID-19 was ruled out in all of them afterwards.

A cross-sectional survey was conducted to have in-depth questionnaire interview of these people who were all uninfected by SARS-CoV-2 in a high-risk area, Hubei province, China. This study aims to identify the common grounds and personal behavior leading to a zero-infection rate among participants that might provide crucial hints on global COVID-19 pandemic control.

# MATERIALS AND METHODS

## Participants, designs and settings

A citizen who presented with body temperature equal to or greater than 37.5 degree Celsius in Hubei province was not allowed for boarding. After arrival to Macao, all 57 citizens were sent to public health clinical center for a 14-day quarantine. A total of three serial nasopharyngeal swabs were obtained on day 2, day 7 and day 13 for viral RNA detection by real-time RT-PCR techniques, which were all negative (100%) (*Macao SAR Government Portal, 2020a*). Sera antibodies of SAR-CoV-2 were tested with all negative results (100%) on day 14 before citizens released from quarantine. All citizens did not complain any symptoms during quarantine period. A questionnaire was designed to obtain demographic information, activity in Hubei province, contact history, personal health behaviors such as habit of handwashing, mask usage and home cleaning. Participants aged 15 or over were eligible for this study. The questionnaire survey was delivered to the isolation ward and was implemented by self-administration. The written consents were collected as digital format.

Infants and children with age under fifteen were considered ineligible for this survey. This study was approved by the Hospital Medical Ethical Committee of Centro Hospitalar Conde de São Januário, Macao SAR, China.

## Statistical method

Descriptive statistic was used to summarize demographic information, high-risk activities and common preventive measures via standard parameter such as percentage, mean and median. Then we compared behavior changes before and during COVID-19 outbreak using Wilcoxon signed rank test in continuous variables or McNemar test in dichotomized variables. The statistical significance level was determined at $\alpha = 0.05$. The statistical analysis was conducted using R (version 3.5.2, *R Core Team, 2018*).

## RESULTS

A total of 42 effective questionnaires were analyzed in final after exclusion of 14 infants and children with age less than 15 years old and missing of one questionnaire (response rate: 97.7%). The demographic information was summarized in Table 1. The majority of the participants aged between 20 and 44 years old (52.4%) and had received secondary education or above (97.6%). The proportion of female composed more than 70% of this group of returners. The most common comorbid diseases were hypertension (7.1%), followed by diabetes mellitus (4.8%) and hepatitis (4.8%). More than half of the respondents were non-smokers (61.9%). The main reason for visit Hubei is to visit relatives (88.1%). More than 85 percent of participants thought the most important reason of not getting COVID-19 was to keep distance ("stay away") from the crowd and decrease cluster or gathering incidence, followed by good personal protective measures (73.8%).

Mobility and participation of high-risk activities were restricted for these participants in Hubei province according to the emergency response policy and these were specified by these respondents (Table 2): 97.6% of them did not visit crowded places; 90.5% of them did not use any public transportation; 90.5% did not go to any supermarket. About three-quarters of respondents received daily supply at home via unified delivery. None of them visited or traveled to other provinces or cities (0%). All the participants (100%) denied any contact with suspected or confirmed COVID-19 patients while 4.8% of the participants stated there was confirmed COVID-19 cases in their local community.

A further survey of comparison of personal preventive measures before and during disease outbreak showed increased alert and practice of personal protection and hygiene during the spread (Table 3), such as wearing a mask when outdoor (16.7% and 95.2%, $P < 0.001$), wearing a mask every time when contact or talk with people (10% and 95%, $P < 0.001$), often wash hands with soap/liquid soap (85.7% and 100%, $P = 0.031$), use of alcohol-based hand sanitizers or disinfected wipes as substitute if handwashing facility not available (71.4% and 95.2%, $P = 0.006$), cleaning clothes and personal belongings immediately once get back home (35.7% and 78.6%, $P < 0.001$), cleaning mobile

**Table 1 Demographic information among the respondents returned from Hubei, China during COVID-19 outbreak in early 2020.**

|  | Total (N = 42) | Male (N = 11) | Female (N = 31) |
|---|---|---|---|
| Proportion | 100% | 26.2% | 73.8% |
| Age (mean, SD) | 40.2, 14.8 | 42.9, 16.8 | 39.3, 14.1 |
| 15–19 | 4 (9.5%) | 1 (9.1%) | 3 (9.7%) |
| 20–44 | 22 (52.4%) | 4 (36.4%) | 18 (58.1%) |
| 45–54 | 10 (23.8%) | 4 (36.4%) | 6 (19.4%) |
| 55–64 | 4 (9.5%) | 1 (9.1%) | 3 (9.7%) |
| 65 or above | 2 (4.8%) | 1 (9.1%) | 1 (3.2%) |
| Education level |  |  |  |
| Primary education or less | 1 (2.4%) | 0 (0%) | 1 (3.6%) |
| Secondary education | 18 (42.9%) | 4 (36.4%) | 14 (50.9%) |
| Bachelor's degree | 16 (38.1%) | 4 (36.4%) | 12 (42.9%) |
| Master's degree or above | 2 (4.8%) | 1 (9.1%) | 1 (18.9%) |
| Missing data | 5 (11.9%) | 2 (18.2%) | 3 (9.7%) |
| Presence of chronic disease(s) | 7 (16.7%) | 2 (18.2%) | 5 (16.1%) |
| Hypertension | 3 (7.1%) | 1 (9.1%) | 2 (6.5%) |
| Diabetes Mellitus | 2 (4.8%) | 0 (0%) | 2 (6.5%) |
| Hepatitis | 2 (4.8%) | 0 (0%) | 2 (6.5%) |
| Dyslipidemia | 1 (2.4%) | 1 (9.1%) | 0 (0%) |
| None | 36 (85.7%) | 10 (90.1%) | 26 (83.9%) |
| Tabagism |  |  |  |
| Current Smoker | 10 (23.8%) | 4 (36.4%) | 6 (19.3%) |
| Ex-smoker | 6 (14.3%) | 2 (18.2%) | 4 (12.9%) |
| Never | 26 (61.9%) | 5 (45.5%) | 21 (67.7%) |
| Reason for visit Hubei |  |  |  |
| Visit relatives | 37 (88.1%) | 10 (90.1%) | 27 (87.1%) |
| Business trip | 0 (0%) | 0 (0%) | 0 (0%) |
| Other reasons | 5 (11.9%) | 1 (9.1%) | 4 (12.9%) |
| Location of accommodation |  |  |  |
| City center | 14 (33.3%) | 4 (36.4%) | 10 (32.2%) |
| Countryside or town | 11 (26.2%) | 3 (27.3%) | 8 (25.8%) |
| Village | 17 (40.5%) | 4 (36.4%) | 13 (41.9%) |
| Number of people cohabit to share the bedroom |  |  |  |
| Alone | 14 (33.3%) | 3 (27.3%) | 11 (35.5%) |
| 2 people | 16 (38.1%) | 3 (27.3%) | 13 (41.9%) |
| 3 people | 8 (19%) | 3 (27.3%) | 5 (16.1%) |
| 4 people or above | 4 (9.5%) | 2 (18.2%) | 2 (6.5%) |

phone regularly (43.9% and 65.9%, $P = 0.012$). Only 11.9% of respondents attend meal gatherings regularly during the spread compared to 59.5% before ($P < 0.001$). The increase in personal measures is significant and may possibly reflect the effectiveness of public health interventions.

**Table 2 High-risk activities and daily supply conditions among respondents during COVID-19 outbreak in early 2020 in Hubei, China.**

| | Participants replied "yes" | | |
| --- | --- | --- | --- |
| | Total (N = 42) | Male (N = 11) | Female (N = 31) |
| Visit other cities in Hubei province (%) | 0 (0) | 0 (0) | 0 (0) |
| Visit other provinces (%) | 0 (0) | 0 (0) | 0 (0) |
| Go to hospital or clinic (%) | 5 (11.9) | 2 (18.2) | 3 (9.7) |
| Contact with suspected/confirmed COVID-19 patients (%) | 0 (0) | 0 (0) | 0 (0) |
| Local community with confirmed COVID-19 cases (%) | 2 (4.8) | 2 (18.2) | 0 (0) |
| Use of public transportation (%) | 4 (8.5) | 2 (18.2) | 2 (6.5) |
| Go to supermarket (%) | 4 (8.5) | 2 (18.2) | 2 (6.5) |
| Visit crowded places (%) | 1 (2.4) | 1 (9.1) | 0 (0) |
| Contact with live poultry (%) | 2 (4.8) | 1 (9.1) | 1 (3.2) |
| Eat game meats (%) | 0 (0) | 0 (0) | 0 (0) |
| Receive daily supply via the unified delivery (%) | 32 (76.2) | 8 (72.7) | 24 (77.4) |

## DISCUSSION

The aims of this research was to investigate the reasons that contributed to the negativeness of COVID-19 in this high-risk population in Hubei province. On the one hand, good physical health could be one factor, as the majority of participants were below the age of 45 (61.9%), non-smokers (61%) and 85.7% had no underlying chronic diseases. However, further studies are needed to determine the exact effect of physical health on the risk of COVID-19 infection. On the other hand, it was also important to stop the transmission chain via political measures or personal health behaviors.

On 23 January 2020 (2 days before the Chinese New Year), the China government imposed a "lockdown" in Wuhan and other cities in Hubei to quarantine this center, which is commonly referred to as the "Wuhan lockdown" (*Health-Commission, 2020*). All public transport, including buses, railways, flights and ferry services were suspended with all stations and airports closed. The residents of Wuhan were not allowed to leave the city without permission which was unprecedented in public health history. Besides, measures on social aspects including the ban on massive gatherings such as concerts or competitions, close of entertainment venues and public facilities, schools closure and mandatory orders of wearing masks in public areas, were applied to mitigate the outbreak by controlling the source of infection and block transmission routes (*Pan et al., 2020*). As a result, the respondents of our study reported the highly restricted mobility in Wuhan, China. A total of 97.6% of them denied visiting crowded places which required high self-discipline and other public measures to cooperate. To achieve this level of mobility restriction, local authority organized a team of volunteers to facilitate the delivery of foods and other supply to each home quarantine family (*Chinanews.com, 2020*), 76.2% of participants received essential materials via this method that decreased the chance of outdoor activity and interaction with other. Nonetheless, 85% of respondents said that
**Table 3 Comparison of personal preventive measures before and during the COVID-19 outbreak among non-infected respondents who traveled to Hubei, China during COVID-19 outbreak in 2020.**

| | Number of missing data | Before outbreak (In Macau) | During outbreak (In Hubei) | P value |
|---|---|---|---|---|
| Wear a mask when outdoor (Yes, %) | 0 | 7/42 (16.7) | 40/42 (95.2) | <0.001* |
| Wear a mask when contact/talk with people (except those living together) | | | | |
| Every time (Yes, %) | 2 | 4/40 (10) | 38/40 (95) | <0.001* |
| Occasional (Yes, %) | 2 | 4/40 (10) | 1/40 (2.5) | 0.375 |
| No (Yes, %) | 2 | 32/40 (80) | 1/40 (2.5) | <0.001* |
| Often wash your hands with soap / liquid soap (Yes, %) | 0 | 36/42 (85.7) | 42/42 (100) | 0.031* |
| If there was no handwashing facility on-site, would you wash your hands with alcohol-based hand sanitizers or disinfected wipes? (Yes, %) | 0 | 30/42 (71.4) | 40/42 (95.2) | 0.006* |
| Clean and disinfect house regularly (Yes, %) | 0 | 36/42 (85.7) | 31/42 (73.8) | 0.227 |
| Frequency of household cleaning and disinfection | | | | |
| No regular cleaning (Yes, %) | 2 | 6/40 (15) | 11/40 (27.5) | 0.227 |
| Once a day (Yes, %) | 2 | 7/40 (17.5) | 13/40 (32.5) | 0.146 |
| Once every 2-3 days (Yes, %) | 2 | 9/40 (22.5) | 10/40 (25) | 1.000 |
| Once a week or more than a week (Yes, %) | 2 | 18/40 (45) | 6/40 (15) | 0.012* |
| Clean mobile phone regularly (Yes, %) | 1 | 18/41 (43.9) | 27/41 (65.9) | 0.012* |
| Clean clothes and personal belongings immediately once get back home (Yes, %) | 0 | 15/42 (35.7) | 33/42 (78.6) | <0.001* |
| Attend meal gatherings (except for family members cohabit with) regularly (Yes, %) | 0 | 25/42 (59.5%) | 5/42 (11.9%) | <0.001* |
| Number of meal gatherings per month (Mean) | 0 | 2.78 | 0.14 | – |
| Number of meal gatherings per month (Median) | 0 | 1 | 0 | <0.001* |

**Note:**
* Indicates P value < 0.05.

"staying away from crowds" was the major reason to be not infected. Moreover, there were emerging evidence suggesting these "lockdown" measures had certain roles on decreasing COVID-19 incidence (*Colbourn, 2020*; *Gostin & Wiley, 2020*; *Klompas et al., 2020*; *Phelan, Katz & Gostin, 2020*; *The Lancet Respiratory Medicine, 2020*). It was estimated that the Wuhan travel ban delayed the epidemic progression by 3–5 days in mainland China, (*Chinazzi et al., 2020*; *Tian et al., 2020*) while reducing case importations to other countries by nearly 80% through mid-February (*Chinazzi et al., 2020*). Furthermore, the rates of confirmed cases and the effective reproduction number (Rt), that is, the mean number of secondary cases generated by a typical primary case at time t in a population, declined since 24 January 2020, and fell below 1.0 since 6 February 2020, in a recent investigation (*Pan et al., 2020*).

Although intensive physical distancing and "lockdown" could help "flattening the curve" on COVID-19 and preventing the sharp upward demand of health system capacities, the consideration of social and economic effects of "lockdown" and knock-on effects on health such as mental health and interpersonal violence is necessary (*Parmet & Sinha, 2020*). Yet, our data showed that over half of the participants (57.1%) felt "calm" during stay in Hubei province, which was somehow counterintuitive. We hypothesized that the provision of sufficient logistic support to the isolated families by local authorities
and clear information delivery to the public during a "lockdown" will help to ease the stress and minimize subsequent psychological impact (*Brooks et al., 2020*). Therefore, local governments should be advised to create a comprehensive strategy and to prudentially evaluate the following concerns including racisms, adequate explanations to the public about the rationale and upside, logistic power and resources, and cultural factors which may hinder the compliance before implementing large-scale mobility restrictions (*Parmet & Sinha, 2020*). The administration of "lockdown" could even lead to precarious situations that could heighten transmission in some countries if corresponding supports are not tailor-made and comprehensive based on their own economic and social conditions, such as workers may be packed in state-run shelter during India "lockdown" (*Pulla, 2020*). Likewise, the announcement of closing the gambling industry during the first half of February in Macau was accompanied with foreseen policies of financial and resources supply could be one of the references of administration of any kind of measures (*Macao SAR Government Portal, 2020c*).

Additionally, the significant behavior changes among participants before and during outbreak consisted of more wearing a mask outdoor, wash hands more frequently, clean and disinfect home more frequently, and less meal gatherings. Although the transmission of SARS-CoV-2 was commonly believed via droplet and contact, no evidence of wearing a surgical mask alone by healthy persons can prevent them from infection with respiratory viruses including COVID-19 currently while inappropriate use/disposal may even increase risk (*World Health Organization (WHO), 2020a*). However, none of the participants in our study agreed that it was less important to wash hands after wearing masks, and all of them (100%) believed that the incidence of accidental touching the face or nose after wearing a mask would be reduced. The effectiveness of personal protective measures in preventing pandemic influenza transmission by meta-analysis showed a significant protective effect of hand hygiene but mixed results for mask use and thus wearing mask was suggested to be applied alongside with hands hygiene (*Saunders-Hastings et al., 2017*). Wearing mask might also act as a "symbolism" on increasing individual awareness of good hygiene practice (*Klompas et al., 2020*). However, the universal mask-wearing scheme in public should be emphasized on the concurrent hand hygiene practice and social distancing as a bundle, while the allocation and availability of resources should be taken into account first to ensure adequate protection for healthcare workers (*Emanuel et al., 2020*).

There were some limitations in this study. The E.L.I.Z.A kits used for antibody detection were qualitative and not able to provide titers information. Although the sample size of this questionnaire was limited and recall bias was inevitable, its implication may indirectly reflect the effectiveness of public health interventions in Wuhan, China, including sanitary cordon, traffic restriction, social distancing, home confinement, centralized quarantine and universal symptom survey. Such interventions were aimed at preventing individuals from face-to-face interaction and preventing asymptomatic COVID-19 patients from spreading the coronavirus within the community. The lack of infected citizens limits for further comparison of difference of measures or behavior and further studies are warranted to determine the effectiveness of each preventive measure on COVID-19 at the

individual level. Moreover, some of the participants had stayed in their relative home where the cleaning duty was not their responsibilities. Hence the question of home cleaning might partially reflect the attitudes from their relatives/friends.

## CONCLUSIONS

Our findings were in line with common preventive measures advised by the World Health Organization. Good personal hygiene and adequate preventive measures such as less gathering, frequent handwashing, in addition to wearing a mask outdoor, were common grounds among 42 uninfected participants during the stay in Hubei province under COVID-19 outbreak. Furthermore, the success of the "lockdown" and self-quarantine policy in Hubei province could contribute to the local authority's strong logistical provision and transparency of information about the policy's rationale in order to maintain better mental health and thus increase compliance and efficacy of preventive measures.

## ACKNOWLEDGEMENTS

We thank Dr. Tan Fong Cheong and Ms. Hong Lei Lou for their assistance in data collection and coordination.

### Funding
The authors received no funding for this work.

### Competing Interests
The authors declare that they have no competing interests.

### Author Contributions
- Chon Fu Lio conceived and designed the experiments, performed the experiments, analyzed the data, prepared figures and/or tables, authored or reviewed drafts of the paper, and approved the final draft.
- Hou Hon Cheong conceived and designed the experiments, performed the experiments, analyzed the data, prepared figures and/or tables, authored or reviewed drafts of the paper, and approved the final draft.
- Chin Ion Lei conceived and designed the experiments, analyzed the data, authored or reviewed drafts of the paper, and approved the final draft.
- Iek Long Lo conceived and designed the experiments, analyzed the data, prepared figures and/or tables, authored or reviewed drafts of the paper, and approved the final draft.
- Lan Yao conceived and designed the experiments, performed the experiments, analyzed the data, prepared figures and/or tables, authored or reviewed drafts of the paper, and approved the final draft.
- Chong Lam conceived and designed the experiments, analyzed the data, authored or reviewed drafts of the paper, and approved the final draft.

- Iek Hou Leong conceived and designed the experiments, analyzed the data, prepared figures and/or tables, authored or reviewed drafts of the paper, and approved the final draft.

## Human Ethics

The following information was supplied relating to ethical approvals (i.e., approving body and any reference numbers):

The Medical Ethical Committee of Centro Hospitalar Conde de São Januário, Macau SAR, China, granted Ethical approval to carry out the study within its facilities.

## Data Availability

The raw measurements are available in the Supplemental File.

## Supplemental Information

Supplemental information for this article can be found online at http://dx.doi.org/10.7717/peerj.9428#supplemental-information.

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
