# Peer review of "The common personal behavior and preventive measures among 42 uninfected travelers from the Hubei province, China during COVID-19 outbreak: a cross-sectional survey in Macao SAR, China"

_PeerJ, doi:10.7717/peerj.9428_

## Round 0.1 · original submission · Minor Revisions

Thank you for raising this important topic in the journal. The reviewers have some critical remarks, on the table organization and the conclusion. Waiting your revised manuscript.

Reviewer 1 ·

Basic reporting

No comment

Experimental design

No comment

Validity of the findings

No comment

Additional comments

This manuscript is addressing an interesting question: to identify common characteristics of personal behavior and preventive measures of 42 uninfected travelers from the Hubei province, China during COVID-19 outbreak. Further, the authors perform a comparison of personal preventive measures before and during disease outbreak, showing increased alert and practice of personal protection and hygiene during the spread of SARS CoV 2. The result will contribute to the improvement of epidemic prevention mesures policies in future and individual health guidelines.

Minor comments:

1. Were IgM and IgG titers quantified on travellers? If yes, please add the information to the text.

2. line 76. "After arrival to Macao, all 57 citizens were sent to public health clinical center for a 14-day quarantine. A total of three serial nasopharyngeal swabs were obtained on day 2, day 7 and day 13 for viral RNA detection by real-time RT-PCR techniques, which were all negative (100%). Sera antibodies of SAR-CoV-2 were tested with all negative results (100%) on day 14 before citizens released from quarantine (6)." This reference do not mention serological tests as indicated on text.

3. All tables needs to be revised. To reduce the excessive unnecessary informations on tables, remove % symbol from first column of all tables and number 42 from all results of table 1 (it is redundant once the number is at the topic Proportion (n=42).

4. Table titles should be presented once. The title is rewrited inside the tables.

5. Table 1. Once sex/gender may influence on adoption of prevention measure and epidemiological data, please organize the second column in male and female

6. Table 1. "Presence of chronic disease(s)" put the none % at the result

7. Table 1. turns "current smoker" a subtopic of "Tabagism" as the "ex smoker" and "never"

8. Table 2. Remove "No" column and reorganize "yes" data on male and female

9. Table 3. set the * as statistics significance at the table footer. It is not clear the P value numbers.

10. line 97 "A total of 42 effective questionnaires were analyzed in final after exclusion of 14 infants and
98 children with age less than 15 years old and missing of one questionnaire (response rate: 97.7%)". It is not clear why some topics at table 3 has only 40 or 41 instead 42 total N?

11. Some numbers of table results are not discussed. The discussion is attentive mainly to lockdown measures and the paper has demografic data (table 1) such as tabagism, chronic diseases, education that may reflet the ausence of COVID19 positive among these group.

·

Basic reporting

The article meets the standarts.

Experimental design

The article meets the standarts.

Validity of the findings

I believe that in the "conclusion" the last sentence (line 201) will look better if it is written clearer, more understandable and highlighter.
Thanks.

Additional comments

Thank you

Reviewer 3 ·

Basic reporting

There are several areas where the authors need to improve writing. For instance, line-81, line-176, line-186, etc.

Experimental design

More details are required for the survey design and its implementation.

Validity of the findings

In general, the findings seem to correspond to intuition. However, in Table 3, why "clean and disinfect house regularly" drops from 36 to 31 for the before and after cases. Is there any specific reason behind this?

---

## Round 0.2 · accepted · Accept

The reviewers have no science remarks. However, I recommend checking the text again, to improve the English language presentation. I believe it does not need another review round and could be done at the final text submission. Thank you again for raising such an important topic on COVID-19 preventive measures.

Reviewer 1 ·

Basic reporting

No comments

Experimental design

NO COMMENTS

Validity of the findings

No comments

Additional comments

The authors clarified all questions and made the suggested changes to improve the text quality.

Reviewer 3 ·

Basic reporting

no comment

Experimental design

no comment

Validity of the findings

no comment

Additional comments

Perform another round of proof-reading and grammatical check.